# Learning Embeddings for Discrete Tree-Structured Data via Structural Prediction

## Abstract

Tree-structured data in natural language syntax, program analysis, and other symbolic domains are typically *discrete, rooted, and ordered* combinatorial objects. Despite their ubiquity, scalable and learnable representations for comparing such discrete structural trees remain limited. Classical methods such as tree edit distance (TED) and tree kernels provide principled structural measures but are computationally prohibitive, while previous neural encoders often produce latent representations without defining a consistent or interpretable space.

We introduce a framework for *learning embeddings for discrete tree-structured data* in which a Transformer encoder is trained through structural prediction tasks—predicting parent indices, node positions, and optionally tree-level categories. Rather than supervising distances directly, these structural objectives induce a coherent Euclidean embedding space for rooted, ordered trees.

A key property of the resulting embedding space is its *stability under local structural perturbations*: a bounded number of edits, such as inserting or deleting a leaf node, produces a proportionally bounded change in the embedding. Empirically, real datasets exhibit a global envelope in which the ratio between embedding distance and edit count remains uniformly bounded. This yields a smoother and more robust structure than TED and other discrete comparison methods, which often exhibit abrupt jumps under minor structural variations.

We demonstrate the effectiveness of our approach across Universal Dependencies treebanks, synthetic random trees, and abstract syntax trees. The learned embeddings correlate strongly with TED, reveal cross-linguistic and cross-parser structural patterns, separate natural from random syntax, and support structure-only code clone retrieval. Together, these results show that structural prediction alone can induce a stable, scalable, and domain-general embedding space that captures fine-grained properties of discrete tree structure.

## 1 Introduction

Tree–structured representations are central to many areas of machine learning, particularly in domains where data naturally take the form of *discrete, rooted, ordered symbolic trees.* Examples include the abstract syntax trees (ASTs) of programs (Mou et al., 2016), hierarchical representations in program analysis (Li et al., 2022), and the dependency and constituency structures of natural language (Nivre et al., 2020). Despite their ubiquity, comparing large collections of such discrete trees remains challenging. Classical distances such as tree edit distance (TED) (Zhang & Shasha, 1989; Pawlik & Augsten, 2016) provide principled structural comparisons but are computationally expensive, even with optimized algorithms such as APTED (Pawlik & Augsten, 2016). Neural encoders such as TreeLSTMs (Tai et al., 2015) and tree-aware Transformers (Shiv & Quirk, 2019) offer expressive latent representations, yet they typically do not yield a calibrated or interpretable embedding space in which distance reflects structural variation in a stable manner.

We propose a general framework for *learning embeddings for discrete tree-structured data* based solely on structural prediction. A Transformer encoder (Vaswani et al., 2017) maps each tree to a fixed-dimensional vector, and Euclidean distances between embeddings define a space for rooted, ordered trees. Crucially, these

Table 1: Comparison of metric properties, supervision requirements, and computational complexity for tree distance and tree representation methods.

| Method | Explicit metric | Vector embeddings | Supervision type | Distance complexity |
|---|---|---|---|---|
| Tree Edit Distance | yes | no | none | $O(n^3)$ |
| Learned TED | yes | no | metric | $O(n^3)$ |
| Tree kernels | yes (implicit) | no | task | $O(n^2)+$ |
| Gromov–Wasserstein | yes | no | unsupervised (OT) | high (OT loops) |
| PQ-grams | yes | no | none / light tuning | $O(n \log n)$ |
| TreeLSTM | no | yes | task | $O(H)$ |
| TreeTransformer | no | yes | task | $O(H)$ |
| GCN | no | yes | task | $O(H)$ |
| Graph2Vec | no | yes | unsupervised | $O(H)$ |
| Ours | yes (Eucl.) | yes | structure | $O(H)$ |

embeddings are *not* supervised by TED or by any gold similarity scores. Instead, they arise implicitly from self-supervised structural objectives, requiring the encoder to predict hierarchical and positional properties of each node, optionally augmented with tree-level labels. This design allows the embedding space to capture structural regularities without imposing an externally specified metric.

A central property of the resulting embedding space is its *stability*. Because the structural features used by our encoder change only locally under common tree modifications—including the insertion or deletion of a leaf—the induced embedding distances respond in a controlled and proportional manner to such edits. A small number of local structural alterations produces a correspondingly small change in the embedding. Empirically, we confirm this behavior by systematically applying controlled edits to trees and observing that embedding distances grow smoothly with the number of modifications, forming a globally bounded and stable envelope. This stands in contrast to the inherently discrete nature of edit-based distances such as TED, where even minor structural differences or annotation inconsistencies can alter the optimal edit sequence and produce disproportionate changes in distance. The learned embedding space therefore provides a smoother and more robust substrate for large-scale structural analysis.

After examining fundamental properties using uniformly random trees, we evaluate the learned embeddings across two structurally diverse domains: natural-language dependency trees and program abstract syntax trees. In natural language, the embedding space reveals cross-linguistic regularities, quantifies parser behavior, distinguishes human annotations from parser and language-model output, and separates natural syntax from synthetic baselines. In code, the same embedding space supports structure-only code clone detection, which we cast as a retrieval problem in the embedding-induced metric space. Clone detection is a standard program-comprehension task for AST-based representations (e.g., (Zhang et al., 2019)), and our embeddings further transfer effectively from natural-language training to program-analysis tasks.

Taken together, these results show that structural prediction alone can induce a coherent, robust, and interpretable space for discrete tree structures, providing a scalable foundation for large-scale structural analysis across heterogeneous domains.

## 2 Related Work

### 2.1 Tree Distances and Learned Structural Metrics

The first block of Table 1 summarizes classical and learned approaches that *directly define or supervise a distance function* over trees. The most widely used combinatorial measure is tree edit distance (TED) (Zhang & Shasha, 1989; Pawlik & Augsten, 2016), which provides a principled notion of structural similarity but requires $O(n^3)$ where $n$ is the tree size, making it impractical for modern corpora. Learned variants attempt

to optimize edit costs or symbol embeddings (Bellet et al., 2012; Paaßen et al., 2018), yet they remain tied to TED-style alignments and thus inherit the cubic-time bottleneck. These methods output a distance but do not yield reusable vector embeddings that support scalable corpus-level or distribution-level analysis.

Other approaches compare trees via implicit similarity functions. Tree kernels (Collins & Duffy, 2001; Moschitti, 2006) compute similarity based on shared substructures but are computationally expensive ($O(n^2)$ or worse), and the resulting kernel does not provide an explicit embedding space. Gromov–Wasserstein (GW) distances (Xu et al., 2019) offer a flexible optimal-transport framework for comparing structured metric spaces, but repeated OT iterations over node sets are often computationally heavy in large-scale pairwise settings. PQ-grams (Augsten et al., 2010; Shindo et al., 2020) provide a fast, locality-sensitive approximation, but their hand-engineered structure limits expressiveness and they provide no guarantees regarding behavior under structured perturbations.

A common limitation across all methods in the first block of Table 1 is that they do not analyze the *sensitivity* of their induced distances to controlled local edits. Because these distances arise from discrete combinatorial operations, they often exhibit abrupt jumps under small structural changes or annotation noise, preventing the formation of a stable geometric space. Our framework addresses this gap by learning a Euclidean embedding space whose behavior under bounded structural modifications is smooth. Unlike prior distance-supervised approaches, our embeddings arise solely from structural prediction objectives and thus provide a continuous, stable substrate for large- scale comparison of discrete tree structures.

## 2.2   Neural Tree Encoders and Structured Representation Learning

The second block of Table 1 consists of neural architectures that obtain tree representations as a *by-product of predictive modeling*, rather than by supervising distances directly. TreeLSTMs (Tai et al., 2015) compute recursive bottom–up node representations that can be pooled into tree-level embeddings. Sequence-to-sequence models have been extended to structured prediction by linearizing tree outputs—e.g., parse trees or grammar productions— and Vinyals et al. (Vinyals et al., 2015a) noted that sequence models can be sensitive to the chosen linearization order. More recently, tree-based Transformers (Wang et al., 2019) encode hierarchical constraints through specialized positional encodings.

General graph encoders can also be applied to trees. Graph convolutional networks (GCNs) (Kipf & Welling, 2017) and unsupervised graph-level embedding models such as Graph2vec (Narayanan et al., 2017) offer flexible representations, but prior studies show that standard GNNs often struggle to capture hierarchical geometry without explicit structural inductive biases (Zhang et al., 2022), a pattern we also observe empirically. Self-supervised geometric encoders for trees (Zhang et al., 2022) perform well when nodes carry real-valued coordinates, but these assumptions do not apply to the *discrete* combinatorial trees considered in this paper.

Neural encoders in the second block of Table 1 produce expressive latent vectors, but the induced similarities are typically dependent on training objectives, and not anchored in any guarantee of smoothness under local structural modifications. Moreover, because they are trained for prediction rather than distance supervision, they provide no principled connection between embedding geometry and structural variation. In contrast, our approach learns an embedding space *entirely from structural prediction tasks*—without supervising distances and without relying on lexical or node attributes. Because the embedding geometry is shaped by structural constraints, it responds smoothly and predictably to bounded edits, yielding a stable space suitable for corpus-level and distribution-level analysis across diverse domains.

## 2.3   Applications: Natural Language and Program Structure

Tree-structured models have long been central to NLP, from rule-based and statistical parsers to contemporary neural architectures for syntactic analysis. Recent probing work examines whether neural sequence models encode latent syntactic structure (Hewitt & Manning, 2019; Chi et al., 2020; Hall Maudslay et al., 2020), while other studies analyze the syntactic properties of LLM-generated parses (Lin et al., 2023). These lines of work focus on representational or empirical properties of parsers and LLMs, but they do not provide a reusable embedding space over explicit tree objects.

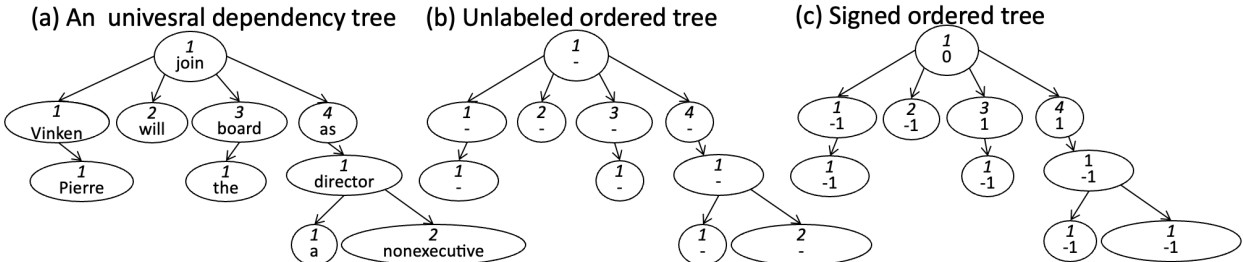

Figure 1: Dependency structure examples of (a) a sentence's dependency tree, and its (b) ordered unlabeled tree and (c) signed ordered tree.

In program analysis, tree structures—particularly abstract syntax trees (ASTs)—are fundamental. Tree-based CNNs (Mou et al., 2016) and recursive neural models capture syntactic patterns for classification and vulnerability detection. ASTNN (Zhang et al., 2019) decomposes large ASTs into statement-level subtrees, while graph-based approaches incorporate control-flow or data-flow edges (Allamanis et al., 2018; Li et al., 2022). Transformer-based models such as GraphCodeBERT (Feng et al., 2020) integrate code tokens with structural signals, and multi-view encoders generalize these ideas to broader program-analysis settings (Wang et al., 2021; Li et al., 2023).

While these models provide strong predictive performance, none gives an explicit, scalable embedding space for comparing *sets of trees*, nor do they analyze how representations change under controlled structural perturbations. In contrast, our framework yields a unified, domain-general embedding space applicable to both linguistic and program structures, enabling stable large-scale analysis of discrete trees.

## 3 Tree-Structured Embedding Space

### 3.1 Discrete Tree-Structured Data and Signed Ordered Trees

Tree-structured data in symbolic domains such as natural-language syntax, program structure, and other discrete hierarchical systems can be formalized as finite rooted ordered trees. Let $\mathcal{T}$ be a set of trees, and $t = (V, E, r, \prec), t \in \mathcal{T}$, where $V$ is the node set, with $n$ as $|V|$, $E \subseteq V \times V$ is the set of directed edges, $r \in V$ is the root, and $\prec$ defines a total order over the children of each node.

Our framework applies to a broad class of discrete tree structures, including dependency trees, constituency trees, and abstract syntax trees (ASTs). While node labels (e.g., words or symbols) can be included, the representation is fundamentally combinatorial. In our setting, we use *signed ordered trees*, where each non-root node $v$ receives a sign $s(v) \in \{-1, +1\}$ indicating whether it lies to the left or right of its parent in a serialization, because our downstream datasets originate from sequential domains such as natural-language text or program code. Nevertheless, the model treats each input strictly as a tree. The linearization (e.g., DFS or BFS traversal) serves only as a deterministic serialization to interface with a sequence-based Transformer and does not encode lexical or surface word-order information. This design ensures that the learned embedding space reflects tree topology rather than properties of the underlying sequence, enabling the same encoder to operate effectively on non-sequential structures such as program ASTs.

Figure 1 illustrates how a dependency tree is converted into an unlabeled ordered tree and then into a signed ordered tree by keeping sibling order and assigning directional signs. The same procedure applies to other symbolic tree structures whenever a linearization is available.

**Node features.** Each node $v$ is associated with a structural feature vector $x_v \in \mathbb{R}^A$. As detailed in Supplementary A, features include DFS index, parent index, depth, subtree sign, arity, sibling index, sentence or program length, and related attributes. Guided by the sequence-to-sequence view of trees in Vinyals et al. (2015b), we represent each tree $t$ by the sequence ordered by DFS index, as follows:

$$X(t) = [\, x_{v_1}, \ldots, x_{v_n} \,] \in \mathbb{R}^{A \times n}.$$

Several alternative linearizations are possible, and sequence models may in general be sensitive to such ordering choices (Vinyals et al., 2015a). In later ablations we evaluate breadth-first (BFS), text-based order, and even random orderings, and find that our method is only mildly affected: despite changes in traversal, the encoder consistently recovers the underlying tree structure.

### 3.2 Learning an Embedding Space for Trees

Our goal is to induce an embedding space for tree shapes *directly from structural prediction signals.* Intuitively, two trees should receive similar embeddings when they exhibit similar structural patterns, regardless of their labels or origins. This leads us to avoid contrastive objectives such as CPC (van den Oord et al., 2018), which may artificially force trees from the same corpus to cluster tightly and thereby obscure meaningful structural variation.

Likewise, we do not supervise the embedding space using a discrete reference measure such as tree edit distance (TED). Because TED is a discrete combinatorial quantity, it does not necessarily reflect the regularities that arise from local order relations, subtree structure, or positional features. For similar reasons, we do not adopt distance-regression approaches such as Siamese models (Bromley et al., 1993; Chopra et al., 2005), which impose an external metric rather than letting the geometry emerge organically from structural constraints.

**Embedding-induced pseudometric.** A pseudometric on $\mathcal{T}$ is a function $d_\theta : \mathcal{T} \times \mathcal{T} \to [0, \infty)$ that satisfies non-negativity, symmetry, the triangle inequality, and $d_\theta(t, t) = 0$. Classical metrics such as TED satisfy these properties but are computationally expensive. Instead, we learn a parametric embedding of dimension $H$:

$$f_\theta : \mathcal{T} \to \mathbb{R}^H,$$

and define a pseudometric on trees by Euclidean distance:

$$d_\theta(t, \tilde{t}) = \|f_\theta(t) - f_\theta(\tilde{t})\|_2. \tag{1}$$

This metric is *induced* by the representation rather than supervised directly.

**Tree encoder with a dedicated representation vector.** Let $N_{\max}$ be the maximum tree size in training. Given feature matrix $X(t) \in \mathbb{R}^{A \times n}$, we prepend a learnable *tree representation vector* (trv) $x_{\mathrm{trv}} \in \mathbb{R}^A$:

$$X'(t) = [\, x_{\mathrm{trv}}, x_{v_1}, \ldots, x_{v_n} \,] \in \mathbb{R}^{A \times (n+1)}.$$

Padding to width $N_{\max} + 1$ yields $\widetilde{X}(t) \in \mathbb{R}^{A \times (N_{\max}+1)}$. A Transformer encoder processes $\widetilde{X}(t)$ columnwise:

$$H_\theta(t) = \Phi_\theta(\widetilde{X}(t)) \in \mathbb{R}^{H \times (N_{\max}+1)}. \tag{2}$$

The embedding is the hidden state of the tree representation vector:

$$f_\theta(t) := H_\theta(t)_{:,1} \in \mathbb{R}^H. \tag{3}$$

Ablations of different serialization (BFS, text-based, random), mentioned previously, always use this trv. Following sentence-level representations in Reimers & Gurevych (2019), we additionally consider a mean-pooling variant as follows:

$$f_\theta^{\mathrm{mean}}(t) = \frac{1}{n+1} \sum_{k=2}^{n+1} H_\theta(t)_{:,k}.$$

**Supervision.** Training data $\mathcal{D}$ consist of quadruples $\{(t^{(i)}, p^{(i)}, q^{(i)}, c^{(i)})\}$, where $k$th node in tree $t^{(i)}$ has observed token position $p_k^{(i)}$, parent index $q_k^{(i)}$ and class $c^{(i)}$. For $k \geq 2$,

$$\pi_\theta(p \mid t, k) = \mathrm{softmax}(W_{\mathrm{pos}} H_\theta(t)_{:,k} + b_{\mathrm{pos}})_p, \tag{4}$$

$$\rho_\theta(q \mid t, k) = \mathrm{softmax}(W_{\mathrm{par}} H_\theta(t)_{:,k} + b_{\mathrm{par}})_q. \tag{5}$$

Tree-level classification is optional:

$$\sigma_\theta(c \mid t) = \mathrm{softmax}(W_{\mathrm{cls}} f_\theta(t) + b_{\mathrm{cls}})_c\,.$$

While not needed to induce a coherent embedding space, class supervision can further align trees at the corpus level (see Supplementary K).

**Loss.** Total loss is defined as follows:

$$\mathcal{L}(\theta) = \mathcal{L}_{\mathrm{predict}}(\theta) + \alpha\,\mathcal{L}_{\mathrm{class}}(\theta),$$

where node-level prediction loss is as follows:

$$\mathcal{L}_{\mathrm{predict}}(\theta) = -\frac{1}{|\mathcal{D}|}\sum_{i=1}^{|\mathcal{D}|}\frac{1}{n^{(i)}}\sum_{k=2}^{n^{(i)}+1}[\log \pi_\theta(p_k^{(i)} \mid t^{(i)}, k) + \log \rho_\theta(q_k^{(i)} \mid t^{(i)}, k)], \tag{6}$$

and tree-level loss as follows:

$$\mathcal{L}_{\mathrm{class}}(\theta) = -\frac{1}{|\mathcal{D}|}\sum_{i=1}^{|\mathcal{D}|}\log \sigma_\theta(c^{(i)} \mid t^{(i)}). \tag{7}$$

During the training phase, each pair $(p_k, q_k)$ in the original matrix A was masked, and the model used the resulting output H to predict the correct values of $(p_k, q_k)$.

### 3.3 Theoretical Considerations of the Induced Geometry

The embedding space is shaped not by distance supervision but by structural prediction objectives. Predicting parent indices encourages sensitivity to hierarchical topology; predicting sequential positions enforces ordering regularities; and optional class prediction aligns representations across corpora. Together, these tasks impose inductive biases that shape $f_\theta(t)$ to reflect the combinatorial structure of discrete trees.

To examine the induced geometry, we analyze how embeddings change under small structural modifications. The representation is constructed via

$$t \mapsto X(t) \mapsto H_\theta(t),$$

where the first column of $H_\theta(t)$ defines $f_\theta(t)$. Prior work derives Lipschitz bounds and sensitivity control mechanisms for attention-based encoders on real-valued sequences (Dasoulas et al., 2021; Qi et al., 2023). While the discrete map $t \mapsto X(t)$ is not naturally Lipschitz under edit operations, we empirically find that $f_\theta(t)$ varies smoothly with the number of local edits.

This yields the following qualitative stability property.

**Proposition 1** (Empirical stability under local structural perturbations). *Let t be a tree and let $\tilde{t}$ be obtained by applying k local structural edits in our perturbation procedure (e.g., leaf insertion/deletion with the same feature update and indexing rules used throughout the paper). Define the embedding-induced distance*

$$d_\theta(t, \tilde{t}) = \|f_\theta(t) - f_\theta(\tilde{t})\|_2.$$

*Across sampled perturbation trajectories, we observe a Lipschitz-like envelope: there exists a constant $L_\theta > 0$ (depending on the encoder parameters and the feature construction/update rules) such that*

$$d_\theta(t, \tilde{t}) \ \leq \ L_\theta \cdot k \quad \textit{(empirically, along sampled trajectories)}. \tag{8}$$

Thus, bounded local structural modifications produce bounded changes in the embedding *in our experiments*. We emphasize that Eq. equation 8 is an empirical stability property rather than a worst-case guarantee over all trees: depending on how structural features are implemented, a single edit may affect feature values beyond a strictly local neighborhood.

In section 5.2, we validate this empirically by applying controlled edits and observing that distances grow smoothly with edit count. In particular, the ratio $d_\theta(t, \tilde{t})/k$ exhibits a stable envelope across sampled trajectories, supporting the claimed Lipschitz-like behavior.

These results show that the embedding space responds smoothly to small structural modifications even without TED supervision. As further shown in section 5.3, $d_\theta$ correlates strongly with TED in practice and preserves local structural neighborhoods, yielding coherent geometry across corpora.

Finally, unlike discrete combinatorial distances such as TED—which can change non-smoothly when a small modification alters an optimal edit sequence—our embedding-induced *pseudometric* often provides a smoother and more interpretable notion of similarity for discrete tree structures in downstream retrieval and visualization.

## 4 Datasets and Settings

We evaluate the proposed method first on random uniform trees, followed by two structurally diverse domains: natural-language dependency trees and abstract syntax trees (ASTs) of source code. These domains differ in branching patterns, depth distributions, and structural motifs, making them complementary testbeds for assessing the generality of the learned tree-structured embedding space.

### 4.1 Natural-Language Treebanks

Our primary experiments use dependency trees from the Universal Dependencies (UD) corpora (Nivre et al., 2020). For each language, we select treebanks that satisfy two conditions: (i) it contains at least 15,000 sentences, allowing stable sampling and training; and (ii) it consists of written, non-translated text. This yields 21 languages including English,[1] shown in detail in Table 7 of Supplementary B. Each UD tree is converted into the signed ordered tree representation described in section 3.1.

To analyze parsers, we additionally construct system-generated treebanks by reparsing the English-EWT corpus using SpaCy, UDPipe, and a parse obtained through prompting a recent large language model ChatGPT-5. These trees allow controlled comparison between human annotations, conventional parsers, and modern generative models (section 6.2).

### 4.2 Random Trees

We generate three classes of synthetic signed ordered trees: (i) basic random trees (uniform, star, balanced, linear, and Markov-generated) (Supplementary C); (ii) context-free random trees, obtained by sampling from a dependency grammar induced from each UD corpus (Supplementary D); and (iii) LLM-generated random trees, derived from syntactic parses of sentences generated by ChatGPT- 2,3,4o,5 in response to SQuAD (Rajpurkar et al., 2016) prompts. For each class of random trees we generate between 10,000 and 100,000 samples, as shown in Table 7.

Among these, the random uniform trees—constructed by attaching nodes uniformly at random to previously generated nodes—are used for basic analysis in section 5.

### 4.3 AST Dataset for Structural Code Similarity

To test whether our framework extends beyond linguistic trees, we consider abstract syntax trees (ASTs) built from Java source code. We follow the standard BigCloneBench (BCB) setting and start from a preprocessed JSONL corpus of Java methods, where each entry provides a unique identifier and the corresponding method body as a code string (`code` or `func` field).

---

[1]The selected treebanks are: English-EWT,Arabic-PADT,Catalan-AnCora,Czech-PDT,Dutch-LassySmall, Estonian-EDT,Finnish-FTB,French-GSD,German-HDT,Hindi-HDTB, Icelandic-IcePaHC,Japanese-GSD,Korean-Kaist,Latvian-LVTB, Norwegian-Bokmaal,Persian-PerDT,Polish-PDB,Romanian-Nonstd, Russian-SynTag,Spanish-Ancora,Turkish-Kenet.

For each method, we parse the code using the Java grammar of Tree-Sitter[2]. The resulting parse tree is converted into a rooted tree-structured representation by traversing only *named* nodes [3], discarding punctuation and other syntactic artifacts.

For every AST we retain a parent array and node order (`head` and `pos_idx`), and discard all lexical content: the encoder sees only tree topology and derived structural features (depth, arity, sibling rank, etc.), not identifiers, literals, or type names. Thus the code experiments evaluate a *purely structural* notion of similarity in program syntax.

### 4.4 Training and Sampling Settings

To train $f_\theta$ (section 3.2), we use AdamW with learning rate $3 \times 10^{-4}$, weight decay $10^{-4}$, batch size 32, and gradient clipping at 1.0. We use cosine learning-rate decay for 10 epochs. All models are implemented in PyTorch and trained on a single GPU.

The embedding dimension is set to $H = 256$ and the class-loss weight is fixed at $\alpha = 0.8$ (Supplementary G). For each dataset domain (UD trees, random trees, ASTs), we sample $K = 10,000$ trees for training (Supplementary E) and $S = 1,000$ held-out trees for validation, unless otherwise noted (Supplementary F). For experiments that require reporting confidence intervals (e.g., correlations and Wasserstein distances), we run $I = 10$ independent trials. TED values are computed on reduced samples (200 trees per class) due to their computational cost.

Models trained on different sets of language classes exhibit stable topological structure, and the learned embedding space interpolates and extrapolates to trees outside the training classes. However, the absolute scale of distances can vary depending on the training set. Therefore, unless stated otherwise, we report results using the model trained on the corpora under examination in each section.

**Visualization and distance computation.** For t-SNE visualizations, we compute the pairwise distance matrix $D_{ij} = d_\theta(t_i, t_j)$ and apply the optimized settings described in Supplementary H. For comparing *sets* of trees (languages, parsers, program families), we use the Wasserstein distance between empirical distributions of embeddings, computed via the Sinkhorn algorithm with entropic regularization (Cuturi, 2013). Class centroids in the 2D maps are shown by large "X" markers.

**Baselines.** We compare against both metric-based and neural baselines, mainly those listed in Table 1. Metric baselines include tree edit distance (TED), tree kernels, Gromov–Wasserstein distance, and PQ-grams, all with standard parameter settings as detailed in Supplementary J.

Neural baselines include TreeLSTM and TreeTransformer, as well as graph-based models GCN and Graph2vec, as summarized in Table 1. Since our method provides a vector representation of sentence tree structure, we also compare with SBERT, computing Euclidean distances between their tree embeddings. Among, Graph2Vec and SBERT are considered only for tree classification, as node-level probes for them would require additional non-trivial procedure. Other neural baselines are trained to predict positions, parents, and optionally classes using the same supervision signals as in section 3.2. We match model sizes (hidden dimension, depth), optimizer, batch size, number of epochs, random seeds, and preprocessing, following Supplementary I. All baselines and our model share identical train/validation splits and sampling protocols, ensuring a fair comparison across domains.

## 5 Fundamental Evaluation of the Embedding Space

We begin by examining the fundamental behavior of the learned embedding-induced distance on uniformly random trees. This provides a controlled setting for evaluating sensitivity, stability, and basic properties before moving to natural-language and program trees.

---

[2] Available through Python package `tree_sitter` (via the `tree_sitter_languages` package).
[3] Nodes where `is_named` is true.

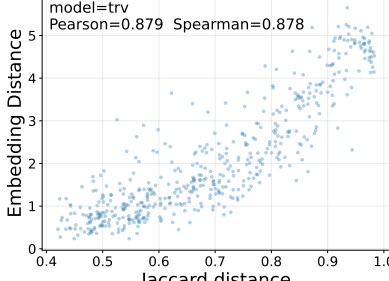 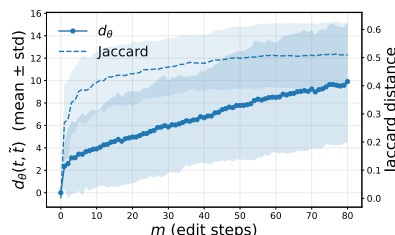 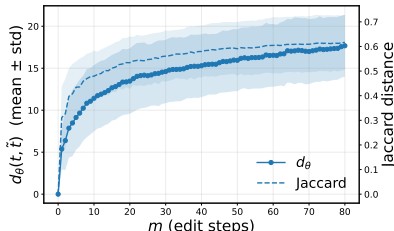

Figure 2: Sensitivity and stability of our learned embedding space. **Left:** Pairwise consistency for 500 pairs of Rand-Uniform trees: $d_\theta(t_i, t_j)$ plotted against their Jaccard distance $J(X(t_i, X(t_j))$, reporting Pearson correlation. **Middle:** Average $d_\theta(t_i, t_j)$ (solid) and $J(X(t_i, X(t_j))$ (dashed) over $m$ random walk trajectories ($m=1,\ldots,80$) random walk trajectories for Rand-Uniform. **Right:** The same experiment on natural-language trees.

For this section, we sample $S = 10000$ trees from the Rand-Uniform treebank and train the encoder using only the structural prediction objectives (parent-index prediction and sequential-position prediction), i.e. without any tree-level classification supervision. Unless otherwise noted, training settings follow section 4.4, with the only difference being that the training corpus consists solely of uniform random trees.

## 5.1 Speed and Scalability

We first compare the computational cost of computing distances among 500 randomly sampled tree pairs obtained by splitting $S = 1000$ instances into halves. The results are summarized in Table 2. Tree edit distance (TED) is the slowest due to its dynamic-programming complexity. An optimized exact variant, APTED (Pawlik & Augsten, 2016), obviously reduces runtime compared to standard TED, but remains orders of magnitude slower than embedding-or kernel-based alternatives. Tree kernels and Gromov–Wasserstein distance are way faster than TED but still incur millisecond-level per-pair costs, which become dominant when the number of comparisons grows quadratic. PQ-grams are faster but still slightly slower than our method.

In contrast, our embedding-induced distance requires computing $f_\theta(t)$ once per tree, after which pairwise comparisons reduce to evaluating an $\ell_2$ norm in $O(H)$ time. This makes the approach scalable to large corpora and enables downstream analyses involving hundreds of thousands of pairwise comparisons.

Table 2: Mean computation time per pair, averaged over 500 tree pairs (ms/pair).

| Method | mean (ms) |
|---|---|
| TED (ZSS) | 352.803 |
| APTED | 141.901 |
| Tree kernels | 1.514 |
| Gromov-Wasserstein | 5.570 |
| PQ-grams | 0.046 |
| Ours | **0.044** |

## 5.2 Local Sensitivity and Empirical Stability

We next examine the embedding space's sensitivity to structural differences and its empirical stability.

**Pairwise sensitivity.** We sample $S = 1000$ random trees, split them into halves, and evaluate 500 pairwise. For each pair $(t_i, t_j)$, we compute (i) the Jaccard distance $J(X(t_i), X(t_j))$ between their feature matrices, and (ii) the embedding distance $d_\theta(t_i, t_j)$. These correspond respectively to the input and output of the encoder (in formula 2).

Figure 2 (left) shows a strong positive trend, with Pearson correlation 0.870. This indicates that the encoder preserves the relative magnitude of structural differences, consistent with the smooth behavior associated with Transformer layers, even though our representation involves additional processing steps (such as the tree representation vector).

**Minimal-edit trajectories.** To probe local behavior, we perform controlled sequences of single-edit modifications. From a random starting tree $t_0$, we apply $M = 80$ successive leaf edits (addition or deletion), while avoiding undo operations and ensuring all intermediate trees differ from previous ones. For each of 30 starting trees, we generate 4 trajectories with different random seeds.

Figure 2 (middle) displays the evolution of $J(X(t_0), X(t_m))$ and $d_\theta(t_0, t_m)$ along $m = 0, \ldots M$. The Jaccard distance rises sharply at small $m$, reflecting its sensitivity to localized edits, and later saturates due to the fixed dimension of $X(t)$. In contrast, the embedding-induced distance responds smoothly: $d_\theta(t_0, t_m)$ increases gradually and becomes approximately linear in $m$. This smoothing reflects the structural constraints internalized by the encoder, consistent with the stability intuition formalized qualitatively in inequality (8).

We note that the standard deviation of $d_\theta(t_0, t_m)$ across random walk trajectories is relatively large. This behavior is expected for two reasons. First, even when the number of edits is fixed, different leaf insertions and deletions affect distinct regions of the tree, producing heterogeneous local structural perturbations. Since the encoder aggregates structural evidence through attention over the entire tree, edits near high-degree or high-depth regions may induce larger representational shifts than edits in shallow or low-branching regions. Second, minimal-edit random walks explore a wide range of tree shapes, and the induced distance therefore reflects both the edit count and the underlying structural diversity of the trajectory. The observed variance thus reflects natural diversity in tree space.

**Natural-language trajectories.** Figure 2 (right) shows the same experiment on natural-language dependency trees. Here, distance curves exhibit a pronounced saturation as $m$ grows, consistent with the empirical distribution of sentence lengths (near log-normal with maximum depth $\approx 13$; see Supplementary C). The leveling-off suggests an effective constant $L$ in equation 8, indicating that the learned embedding space is locally stable for real syntactic trees.

### 5.3 Alignment with Tree Edit Distance

Finally, we examine how well the induced embedding geometry aligns with TED. We compute TED for 500 random pairs drawn from a split of $S = 1000$ trees and compare it with several baselines: (i) exact Collins–Duffy Subset Tree Kernel (SST), (ii) PQ-gram distance with $p=2, q=3$, and (iii) entropic Gromov–Wasserstein distance. We also evaluate our $d_\theta$ and the ablations defined in section 3. Pearson correlations with TED are shown in Table 3.

Our embedding-induced distance shows the strongest correlation with TED (0.75) despite receiving no TED supervision. This indicates that TED-like structural information emerges naturally from the structural prediction tasks. PQ-grams and tree kernels show moderate alignment, while Gromov–Wasserstein exhibits weak correlation.

All ablations remain positively correlated with TED, indicating that TED-aligned structure emerges robustly across variants. Using a plain text order traversal (Ablation-Text-Ordering, 0.69) reduces correlation, suggesting that a structure-aware traversal provides a modest but consistent advantage, likely because it better preserves locality patterns that resemble TED edit operations. Switching the traversal from DFS to BFS yields only a small drop from the full model to 0.73, implying that the metric is relatively insensitive to the exact tree traversal as long as

Table 3: Correlation with TED.

| Method | TED Corr |
|---|---|
| Tree Kernels | 0.56 |
| Gromov–Wasserstein | 0.14 |
| PQ-gram | 0.32 |
| TreeLSTM | 0.26 |
| TreeTransformer | 0.62 |
| GCN | 0.41 |
| Graph2Vec | 0.12 |
| Ablation-Text-Ordering | 0.69 |
| Ablation-BFS | 0.73 |
| Ablation-Random | 0.52 |
| Ablation-Mean-Pool | 0.59 |
| Ours | 0.75 |

it is structure-driven. Ablation-Random still achieves a clear positive correlation (0.52) despite disrupting traversal-induced locality, suggesting that the learned geometry captures a more universal structural signal rather than overfitting to a particular ordering heuristic. Mean-Pool produced reduced result closer to Random. In general, our proposed TRV using DFS ordering produced the best alignment with TED.

Overall, the results confirm that the learned embedding space captures fine-grained structural variation and provides a smooth, stable, and TED-aligned notion of similarity even for uniformly random trees.

# 6 Downstream Evaluation 1: Sentence-Tree Evaluation on UD Corpora

We now demonstrate the usefulness of the learned embedding space on two downstream applications. In this section we focus on natural-language syntax; the next section turns to program structure. Many other uses are possible, including analyses of cross-linguistic variation as illustrated in Supplementary K, but here we concentrate on the most informative examples. Unless otherwise stated, we use the encoder trained with tree-level class supervision on the set of tree datasets under consideration.

## 6.1 Prediction Tasks

We first probe whether the encoder captures syntactic tree structure on UD treebanks using the same supervision signals as in section 3.2. We consider three tasks:

**Sequential position (Seq-ID)** predicts the DFS position index of each node;

**Parent position (Parent-ID)** predicts the index of the parent node; and

**Tree class** predicts the language or corpus identifier from a single tree embedding $f_\theta(t)$.

We evaluate three scenarios:

**S1: In-language**. The task is to distinguish two $S$ sets sampled from the same English-EWT treebank. The ideal language–prediction accuracy is 50%, corresponding to a 50–50 confusion between the two splits.

**S2: English vs. 4 distant languages.** The model is trained on five languages: English-EWT and four typologically distant treebanks (Japanese-GSD, Korean-Kaist, Icelandic-IcePaHC, and Hindi-HDTB). We report the categorization accuracy for distinguishing the five languages.

**S3: All natural-language UD treebanks** that appear in the first block of Table 7.

Accuracies range from 0 to 1; for S1, where the two halves of English-EWT are treated as separate classes, the ideal language prediction accuracy is 50% (a 50–50 confusion between the two splits).

As an ablation, we include **Ablation-Mean-Pool**. As structural baselines, we compare with TreeLSTM, TreeTransformer, and a GCN operating on the dependency tree. These models provide node-level embeddings for Seq-ID and Parent-ID, and embeddings for Tree class. In contrast, as mentioned in section 4.4, Graph2Vec and SBERT yield only graph-level or sentence-level embeddings that are not aligned with UD nodes, and designing node-level probes for them would require additional and non-trivial procedure. We therefore report their scores only for Tree class, where their native outputs are directly comparable.

Table 4: Prediction task accuracy (for S1-language the best is 50%, i.e., a 50–50 confusion ; otherwise, higher is better).

| Scenario | Model | Seq-ID | Parent-ID | Tree Class |
|---|---|---|---|---|
| **S1**:EWT A vs. EWT B | TrLSTM | 12.68% | 32.84% | 48.16% |
| | TrTransformer | 47.81% | 41.51% | 51.25% |
| | GCN | 15.17% | 13.64% | 49.71% |
| | Graph2Vec | - | - | 47.33% |
| | SBERT | - | - | 52.85% |
| | Ours (Mean-Pool) | 45.76% | 53.98% | 49.92% |
| | Ours | **75.91**% | **73.61**% | **50.04**% |
| **S2**:EWT and Far 4 | TrLSTM | 11.68% | 28.72% | 46.20% |
| | TrTransformer | 51.72% | 39.88% | 73.71% |
| | GCN | 14.66% | 13.30% | 57.01% |
| | Graph2Vec | - | - | 43.60% |
| | SBERT | - | - | 32.38% |
| | Ours (Mean-Pool) | 49.16% | 55.06% | 89.17% |
| | Ours | **65.38**% | **64.53**% | **90.68**% |
| **S3**:All 21 NL UDs | TrLSTM | 11.69% | 26.89% | 12.70% |
| | TrTransformer | 54.37% | 64.68% | 41.27% |
| | GCN | 14.15% | 14.02% | 21.55% |
| | Graph2Vec | - | - | 16.48% |
| | SBERT | - | - | 10.55% |
| | Ours (Mean-Pool) | 66.88% | 67.66% | 43.32% |
| | Ours | **82.97**% | **78.01**% | **44.68**% |

Table 4 shows mean accuracies over $I = 10$ trials (standard deviations $\approx 0.03$). In S1 and S2, TreeTransformer performs well on Seq-ID, but our model with a dedicated tree representation vector achieves higher position

and parent accuracy in all settings, with particularly large gains in S3 (82.97% for Seq-ID and 78.01% for Parent-ID). The mean-pooling ablation is consistently weaker on node-level tasks (e.g., 66.88% vs. 82.97% for Seq-ID in S3), suggesting that concentrating structural information in a dedicated tree representation vector (trv section 3) makes Seq-ID and Parent-ID more easily recoverable. For Tree class, the difference between pooling schemes is small: in S2 the two are nearly identical (89.17% vs. 90.68%), and in S3 mean pooling performs competitively (43.32%). Graph2Vec and SBERT achieve reasonable accuracies in S1 (47.33% and 52.85%), but decrease in S2 and S3, confirming that purely graph-level or semantic embeddings do not capture the fine-grained syntactic structure targeted by our probes.

## 6.2 Evaluating Parsers

Parser quality is typically assessed by node/sentence-level accuracies. Here we show that the learned embedding space provides an additional tree-level evaluation signal.

Figure 3 visualizes English-EWT trees together with the same sentences reparsed by UDPipe, SpaCy, and ChatGPT-5. UDPipe lies close to English-EWT, which is unsurprising since it is trained on UD corpora, whereas SpaCy is located further away. ChatGPT-5 is closer than SpaCy: the English-EWT points (red) almost entirely overlap the UDPipe clusters (blue) and ChatGPT-5 (black). On English-EWT, sentence level parsing accuracies are 87.36%, 21.23%, and 76.21% for UDPipe, SpaCy, and ChatGPT-5, respectively, with a random-parser baseline of 16.15%. These values correlate well with mean per-sentence distances in the embedding space: 4.773±0.301, 8.532±0.200, and 4.680±0.190, respectively. This illustrates that our learned embedding space supports parser assessment beyond simple accuracy.

To quantify this effect at scale, we evaluate all three parsers across 13 UD treebanks, i.e., languages for which both UDPipe and SpaCy models are available. For each treebank and each parser, we compute the mean distance between the gold trees and the parser outputs under each metric, then average these means across treebanks.

The first three columns of Table 5 report the resulting mean distances for UDPipe, ChatGPT-5, and SpaCy. Over these 13 languages, the mean accuracy order is UDPipe (0.742) > SpaCy (0.647) > ChatGPT-5 (0.543), which differs from the English-only case because ChatGPT-5 performs worse on non-English languages. In terms of reproducing this ordering in distance space, our embedding-induced distance does so consistently, and the same is true for the learned neural baselines (TreeLSTM, TreeTransformer, and GCN). By contrast, the non-neural distances (TED, tree kernels, Gromov–Wasserstein, and PQ-grams) more often blur these differences and can place

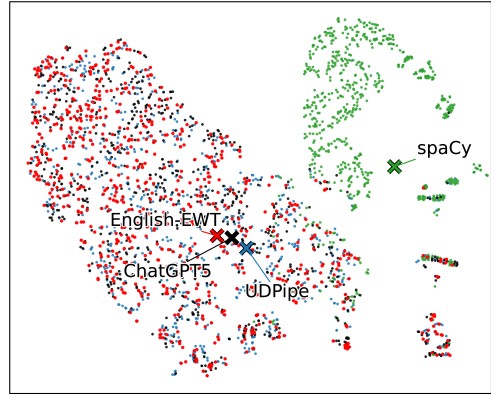

Figure 3: English-EWT (red) and its reparses by UDPipe (blue), SpaCy (green), and ChatGPT-5 (black).

Table 5: Mean distances between gold and parsed trees, and correlation with parser accuracies across 13 UD treebanks. .

|  | Mean Distance | | | Correlation |
| Method | UDPipe | ChatGPT-5 | SpaCy | with parser accuracies |
| --- | --- | --- | --- | --- |
| TED | 4.344 | 8.154 | 3.918 | -0.832 |
| Tree kernels | 4.238 | 6.396 | 4.023 | -0.837 |
| Gromov-Wasserstein | 0.254 | 0.274 | 0.274 | -0.838 |
| PQ-grams | 0.073 | 0.108 | 0.070 | -0.842 |
| TreeLSTM | 7.978 | 15.707 | 9.780 | -0.913 |
| TreeTransformer | 10.002 | 19.935 | 12.239 | -0.911 |
| GCN | 4.139 | 7.200 | 4.911 | -0.818 |
| Ours | 6.544 | 12.945 | 8.298 | **-0.974** |

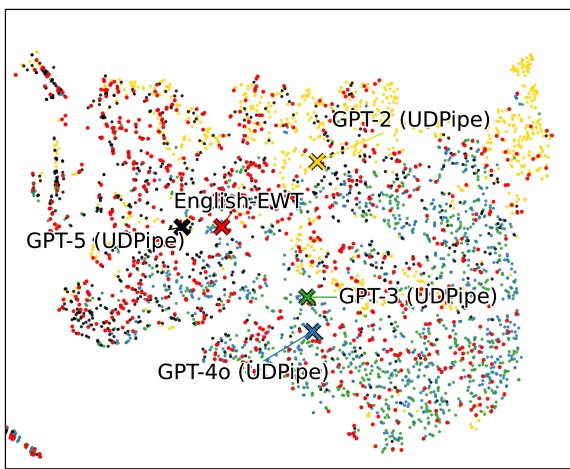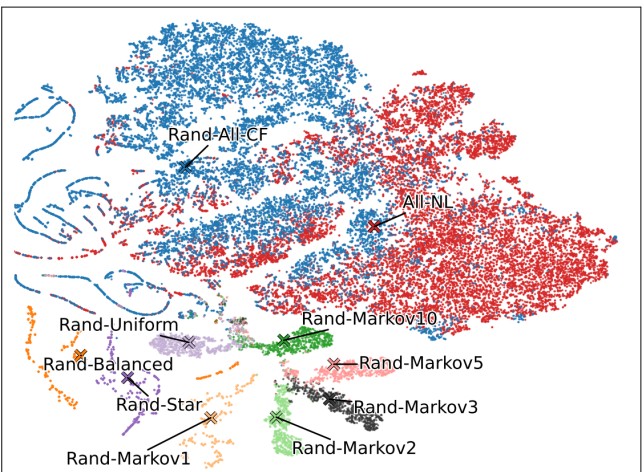

Figure 4: Left: English-EWT and texts generated by ChatGPT- {2 (yellow), 3 (green), 4o (blue), 5 (black)}, reparsed by UDPipe. Right: Embedding space of all natural languages (ALL-NL, red) in Table 7, all context-free trees (Rand-ALL-CF, blue), and other, simpler random trees.

SpaCy closer to the gold trees than UDPipe or swap SpaCy
and ChatGPT-5, suggesting weaker sensitivity to parser-quality differences in this cross-lingual setting.

We further summarize the relationship between distance and accuracy by computing, for each metric, the correlation across the three parsers between their mean distances and their mean accuracies. Since distance should decrease as accuracy increases, the ideal correlation is $-1$. The last column of Table 5 shows that our method achieves the strongest negative correlation $(-0.974)$, closer to the ideal than any baseline, including the neural ones. This indicates that while several learned encoders can recover the coarse accuracy ordering, our induced geometry tracks the parser-quality differences more faithfully, yielding a sharper tree-level notion of "parser closeness" across languages.

## 6.3 Evaluating Tree Structure Quality of Language Models

Language models have traditionally been evaluated using sequence-based metrics such as perplexity (Brown et al., 1992). However, perplexity does not further provide an understanding how well they capture the *underlying tree structure* of language. Here we show how the learned embedding space provides a way to assess the structural quality of model outputs.

In our first example, we parse texts generated by ChatGPT 2–5 (as described in section 4.2) using UD-Pipe, which performed the best in the previous section. The left panel of Figure 4 shows the resulting embedding space. We observe that the Chat-GPT models—especially versions 2 and 3—form clusters that are further from English-EWT. The mean Wasserstein distances from English-EWT (last column of Table 7) show that GPT-generated structures move closer to English-EWT across model generations from 2 to 5. This trend is *not* captured by TED (third column in the last block of Table 7, where TED produces a reversed ordering), highlighting the additional insights offered by our embedding-based distance.

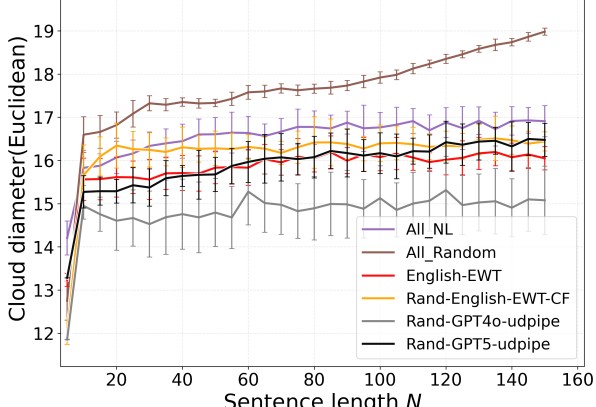

Figure 5: Diameters of tree sets for sentence samples of length $\leq N$.

As a second example, we compare UD trees against (i) basic random trees (uniform, star, balanced, linear, and Markov-generated), and (ii) context-free trees sampled from grammars inferred from each UD corpus. In Figure 4 (right), in all languages, UD trees occupy a region of the embedding space that is well separated

Table 6: Accuracy of distinguishing random and model-generated trees from English-EWT.

| Source | Simple random | Markov | Context-free | GPT-2 | GPT-3 | GPT-4o | GPT-5 |
|---|---|---|---|---|---|---|---|
| Accuracy | 99.88% | 99.77% | 99.27% | 84.06% | 70.89% | 72.67% | 66.67% |

from both random families, indicating that the learned embeddings capture the nontrivial combinatorial constraints of natural syntax.

Figure 5 provides a quantitative summary via the mean diameter of the embedding clouds, defined as the maximum pairwise distance within the set of trees whose sentence length is at most $N$. To ensure comparability across datasets, all distances are computed using a single model trained only on natural-language trees.

Across all datasets, the diameter increases rapidly for short sentences and transitions to a slower growth rate beyond $N \approx 20$. The kink near $N = 20$ reflects the fact that the lengths of random sentences were sampled from an empirical sentence-length distribution (see Figure 7 and Supplementary C). The vertical ordering of the curves matches intuitive structural complexity: `All_Random` shows the largest and fastest-growing diameters, followed by `All_NL`.

For model-generated text, the small diameter of `Rand-GPT4o-udpipe` suggests relatively limited structural variability in GPT-4o outputs. In contrast, `Rand-GPT5-udpipe` exhibits diameters much closer to English-EWT, indicating that GPT-5 produces syntactic structures with greater diversity and complexity. This illustrates how the learned embedding space can reveal structural differences among LLM generations that are not captured by traditional tree-edit distances.

Lastly, Table 6 quantifies how easily trees from each synthetic source can be distinguished from English-EWT based on their embeddings. For simple random trees, Markov trees, and context-free trees, the classifier reaches very high accuracies ($\approx 99\%$), showing that these synthetic families occupy regions of the space that are clearly separated from natural English syntax. In contrast, GPT-generated trees are relatively harder to discriminate from English-EWT: accuracies drop to around 67–84%, only moderately above chance. Among the language models, GPT-5 yields the lowest accuracy (66.67%), i.e., it is the hardest to distinguish from English-EWT, indicating that its syntactic structures are closest to those of real English text. Taken together, these results demonstrate that the learned embedding space is simultaneously sensitive to reject unrealistic random trees and fine-grained to capture gradual structural improvements across LLM generations.

# 7 Downstream Evaluation 2: Structure-Only Code Clone Detection

To assess whether the learned embedding space transfers beyond natural language, we evaluate it on *structure-only* code clone retrieval using abstract syntax trees (ASTs) derived from Java methods in Big-CloneBench (BCB).

**Encoders and embeddings.** We compare three encoders that all share the same architecture and structural prediction objectives introduced earlier.

- *AST-only*: trained solely on Java ASTs.

- *NL-only*: trained only on natural-language dependency trees from UD treebanks and never exposed to code during training; at test time, we directly feed ASTs into this encoder using the same structural feature templates.

- *Mix-Training (AST+UD)*: trained on a mixture of AST and UD trees, to test whether joint supervision across domains improves retrieval.

Clone detection is cast as a retrieval problem in the embedding-induced metric space. Let $\mathcal{U}$ be the set of queries. For each $u \in \mathcal{U}$ we retrieve an *ordered* list of its $k$ nearest neighbors $\mathrm{NN}_k(u) = (v_1, \ldots, v_k)$ among all other queries (excluding $u$ itself), where neighbors are ranked by $d_\theta$. Let $\mathcal{C}(u)$ denote the set of true clones of $u$.

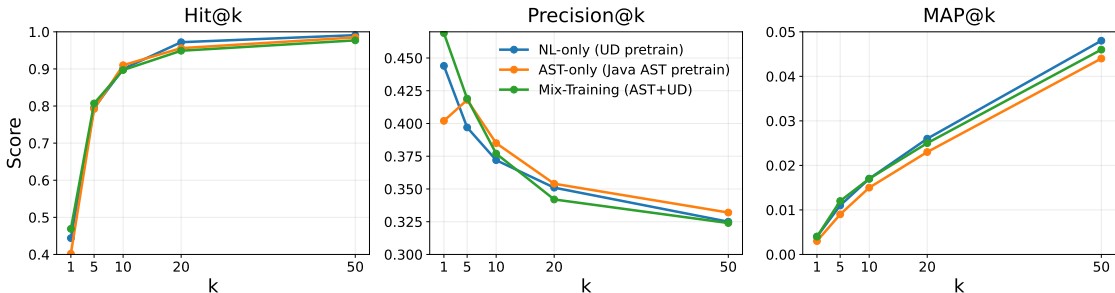

Figure 6: Retrieval performance on structure-only AST clone retrieval. Left: Hit@k. Middle: Precision@k. Right: MAP@k.

We report Hit@$k$,

$$\text{Hit@}k \ = \ \frac{1}{|\mathcal{U}|} \sum_{u \in \mathcal{U}} \mathbf{1}\big[\, \mathcal{C}(u) \cap \{v_1(u), \ldots, v_k(u)\} \neq \emptyset \,\big],$$

the fraction of queries for which the top-$k$ retrieved neighbors contain at least one true clone. Here $v_1(u), v_2(u), \ldots$ are the neighbors of query $u$ ranked by increasing distance (excluding $u$ itself), and $k$ is a fixed cutoff.

To evaluate ranking quality beyond the first hit, we also report Precision@$k$ and MAP@$k$. Precision@$k$ measures how many of the top-$k$ retrieved neighbors are true clones (purity), while MAP@$k$ additionally captures how early those clones appear in the ranking.

For a query $u$, the per-query precision at cutoff $k$ is

$$\text{Prec@}k(u) \ = \ \frac{1}{k} \sum_{j=1}^{k} \mathbf{1}[\, v_j(u) \in \mathcal{C}(u) \,], \qquad \text{Precision@}k \ = \ \frac{1}{|\mathcal{U}|} \sum_{u \in \mathcal{U}} \text{Prec@}k(u).$$

We compute AP@$k$ using truncated average precision:

$$\text{AP@}k(u) \ = \ \frac{1}{|\mathcal{C}(u)|} \sum_{j=1}^{k} \text{Prec@}j(u) \cdot \mathbf{1}[\, v_j(u) \in \mathcal{C}(u) \,], \qquad \text{MAP@}k \ = \ \frac{1}{|\mathcal{U}|} \sum_{u \in \mathcal{U}} \text{AP@}k(u).$$

Queries with $\mathcal{C}(u) = \emptyset$ are excluded from Precision@$k$/MAP@$k$.

**Results: zero-shot transfer and mixed training.** Figure 6 shows that the NL-only encoder, despite never seeing code during training, achieves strong clone retrieval performance on ASTs: Hit@1 is 0.444 and Hit@10 reaches 0.900. For reference, a random ranking baseline yields Hit@1 $\approx$ 0.291 in this setup, indicating that the learned geometry is more informative than chance even without code-specific pretraining.

Comparing NL-only and AST-only, the two models perform similarly across $k$. NL-only is slightly better at small $k$ (Hit@1 and Hit@5), while AST-only is marginally better at Hit@10. Mixed training sharpens the very top of the ranking: Mix-Training attains the best Hit@1 (0.469) and Hit@5 (0.807), as well as the best MAP@5 (0.012). However, the mixed model does not uniformly dominate for larger $k$, where Hit@20/50 and Precision@50 decrease slightly. This suggests that cross-domain training can improve nearest-neighbor quality while modestly reshaping the broader neighborhood structure.

Overall, these results show that a single structure-trained encoder can induce a meaningful embedding geometry over heterogeneous tree domains (dependency trees and program ASTs). In particular, the strong zero-shot performance of NL-only supports the view that the learned embeddings capture domain-general aspects of hierarchical structure, enabling nontrivial code clone retrieval from purely structural information, without any lexical cues.

# 8 Conclusion

We introduced a scalable framework for *learning embeddings for discrete tree-structured data via structural prediction*. Rather than supervising distances directly, the proposed method induces an embedding space whose Euclidean geometry reflects the combinatorial structure of trees. The resulting pseudometric is computationally efficient, stable under local structural perturbations, and suitable for large-scale comparison of tree sets—offering orders-of-magnitude speedups over classical distances such as TED.

Across natural-language dependency trees, the learned embedding space captures cross-linguistic organization, separates human annotations from parser and LLM outputs, and reveals structural trends—such as the progressive syntactic improvement across ChatGPT versions—that are not recoverable from edit-based metrics alone. Experiments with synthetic random trees further demonstrate that natural syntax occupies a region of the space, distinct from trees generated by simple random or grammar-based processes.

The same embedding space transfers effectively beyond natural language. In structure-only AST clone detection, the learned geometry supports competitive retrieval using no lexical cues, and even a model trained solely on natural-language trees performs strongly in zero-shot settings. These results suggest that structurally trained embeddings capture domain-general principles of hierarchical organization, enabling meaningful comparison across heterogeneous tree domains.

More broadly, this work highlights the potential of structural prediction as a foundation for building coherent spaces over discrete combinatorial objects. Future work includes incorporating lexical or semantic information into the embedding space, extending the framework to broader graph families, and developing applications in cross-lingual typology, code analysis, and large-scale structural modeling.

# 9 Limitations

Although the proposed framework provides a scalable and stable embedding space for discrete tree-structured data, several limitations merit discussion.

**Dependence on structural encoding.** The method relies on a fixed set of structural features and a serialization of each tree. While we show that structure-aware traversals (e.g., DFS, BFS) yield broadly similar behavior, the choice of representation may influence performance in subtle ways. Extensions to more flexible or learned structural encodings remain an open direction.

**Lack of lexical or semantic information.** By design, the model ignores all node labels and focuses exclusively on tree topology. This enables cross-domain generalization but also limits applicability in settings where lexical content is essential (e.g., semantic code clone detection, fine-grained linguistic analysis). Integrating lexical or type-level information while preserving the stability of the metric is left to future work.

**Approximate, not guaranteed, stability.** Our analysis provides qualitative arguments and empirical evidence for stability under local structural edits, but does not establish formal Lipschitz guarantees with respect to edit operations. The embedding's behavior under larger or adversarial modifications remains only partially understood.

**Tree-specific design.** The framework is tailored to rooted, ordered trees. Many real-world datasets include richer graph structures such as control-flow graphs, or phylogenetic graphs. Extending the approach to general graphs while maintaining interpretability is non-trivial.

**Training-data dependence.** The absolute scale and global geometry of the embedding space depend on the distribution of trees seen during training. While the metric generalizes across domains to some extent, embeddings trained on one tree family (e.g., UD) may exhibit different inductive biases when applied to another (e.g., ASTs). Understanding how to normalize or compose metrics across domains is an open question.

**Computational considerations.** Although the learned metric is much faster than classical distances at evaluation time, training requires a Transformer encoder and thus incurs non-negligible GPU cost. Very large trees may also require additional padding and memory, though our experiments restrict tree sizes to realistic limits from UD and Java ASTs.

Overall, these limitations suggest several promising avenues for future work, including integrating lexical information, extending structural prediction to richer graph classes, and tightening theoretical characterizations of the induced space.

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

## Supplementary Material

## A   Tree Node Features

Before being fed to the Transformer, each node is represented as a vector of $A$ features, and trees are linearized in depth-first order with a validity mask for padding. We use the following six features identically across all models:

- **DFS node position:** integer in $\{0, \ldots, n-1\}$ indicating the node's DFS order.

- **Node position:** integer in $\{0, \ldots, n-1\}$ indicating the index in the sentence.

- **Parent position:** integer in $\{0, \ldots, n-1\}$ indicating the parent's DFS position in the sentence.

- **Depth:** integer depth of the node from the root.

- **Modifying direction:** categorical feature encoding the attachment direction relative to the parent, with -1 before and +1 after.

- **Arity**: number of node's children.

- **Number of siblings:** number of siblings sharing the same parent.

- **Position among siblings:** order among siblings.

- **Sentence size:** $n$, the tree's length (number of nodes).

- **Position of sentence head :** integer in $\{0, \ldots, n-1\}$ indicating the sentence's head position.

## B   Wasserstein Distances of centroids of natural language related trees from that of English-EWT

Table 7 summarizes the tree datasets used in our evaluation and reports, for each dataset, the mean pairwise distance under (i) tree edit distance (TED) and (ii) the embedding-induced distance learned through structural prediction. The table is organized into three horizontal blocks corresponding to the main domains analyzed in this work: natural-language dependency treebanks, English-EWT sentences re-parsed by different parsers or LLMs, and synthetic random tree families. This organization reflects the cross-domain scope of our embedding framework, allowing direct comparison of structural behavior across linguistic, model-generated, and artificially constructed trees. All TRV distances in this table are computed with the model trained under all natural languages in this block.

**Natural-Language Treebanks (Top Block)**   The first block lists 21 Universal Dependencies (UD) treebanks that meet our size and genre criteria. For each treebank, we report the number of trees, the mean TED and embedding-induced distances from English-EWT. These values reflect structural similarity across languages. The learned distances provide a smoother and more discriminative spectrum than TED, capturing cross-linguistic structural relationships (e.g., Romance vs. Germanic vs. Uralic languages, see Supplementary K, too) even though the encoder has no access to lexical information. This illustrates how the embedding space learned through structural prediction encodes typological structure in a way.

**English-EWT Re-parsed by Parsers and LLMs (Middle Block)**   The second block examines structural consistency across parsing systems by reparsing the same English-EWT sentences with SpaCy, UDPipe, and ChatGPT 4o and 5. Because these treebanks contain identical sentences, differences in distance reflect only the structural discrepancies introduced by the parser or generative model, as was discussed in the main text.

Table 7: Treebanks used for the experiments. We report the number of trees in each treebank and the mean Wasserstein distance from English-EWT (obtained over $I$=10 trials) using TED and our TRV distance. For the TRV column, we use the encoder trained on all natural-language data from the first block.

| treebank | # trees | TED | Ours(TRV) |
|---|---|---|---|
| Natural Language | | | |
| English-EWT | 16621 | 2.984±0.173 | 4.341±0.142 |
| Arabic-PADT | 19738 | 7.087±0.044 | 11.245±0.261 |
| Catalan-AnCora | 16678 | 5.145±0.069 | 10.431±0.299 |
| Czech-PDT | 87913 | 5.557±0.066 | 5.960±0.228 |
| Dutch-LassySmall | 17120 | 5.178±0.046 | 5.811±0.097 |
| Estonian-EDT | 30972 | 5.771±0.051 | 6.247±0.180 |
| Finnish-FTB | 18723 | 6.789±0.034 | 8.914±0.202 |
| French-GSD | 18535 | 5.001±0.070 | 8.643±0.152 |
| German-HDT | 189928 | 5.300±0.099 | 7.306±0.252 |
| Hindi-HDTB | 16647 | 7.300±0.084 | 9.348±0.164 |
| Icelandic-IcePaHC | 44029 | 6.286±0.072 | 8.681±0.292 |
| Japanese-GSD | 57109 | 7.676±0.042 | 10.036±0.183 |
| Korean-Kaist | 27363 | 9.585±0.127 | 10.377±0.179 |
| Latvian-LVTB | 15984 | 5.739±0.080 | 6.100±0.167 |
| Norwegian-Bokmaal | 20044 | 5.101±0.054 | 5.754±0.228 |
| Persian-PerDT | 29107 | 7.038±0.072 | 7.781±0.128 |
| Polish-PDB | 22152 | 5.668±0.091 | 6.636±0.198 |
| Romanian-Nonstd | 26225 | 5.212±0.061 | 7.749±0.176 |
| Russian-SynTag | 87336 | 5.696±0.094 | 6.400±0.082 |
| Spanish-Ancora | 17662 | 5.243±0.070 | 9.702±0.188 |
| Turkish-Kenet | 18687 | 8.647±0.052 | 10.507±0.174 |
| English-EWT re-parsed by parsers | | | |
| English-EWT-Spacy | 16621 | 9.479±0.121 | 8.532±0.200 |
| English-EWT-UDPipe | 16621 | 4.178±0.100 | 4.773±0.301 |
| English-EWT-ChatGPT4o | 16621 | 6.111±0.110 | 6.03±0.194 |
| English-EWT-ChatGPT5 | 16621 | 4.634±0.071 | 4.680±0.190 |
| Random Trees | | | |
| Rand-Uniform | 100000 | 9.390±0.094 | 14.736±0.176 |
| Rand-Star | 100000 | 12.264±0.129 | 16.393±0.271 |
| Rand-Balanced | 100000 | 12.617±0.079 | 15.584±0.166 |
| Rand-Linear (ord=2) | 100000 | 16.581±0.208 | 17.98±0.156 |
| Rand-Markov (o=3) | 100000 | 13.042±0.134 | 17.220±0.173 |
| Rand-Markov (o=5) | 100000 | 11.324±0.103 | 16.391±0.197 |
| Rand-Markov (o=10) | 100000 | 10.167±0.137 | 15.374±0.206 |
| Rand-English-EWT-CF | 100000 | 6.554±0.073 | 11.692±0.159 |
| Rand-GPT2 | 12880 | 5.116±0.065 | 8.375±0.189 |
| Rand-GPT3 | 14327 | 5.169±0.039 | 7.655±0.288 |
| Rand-GPT4o | 19327 | 5.263±0.062 | 7.335±0.210 |
| Rand-GPT5 | 15996 | 5.464±0.065 | 5.881±0.267 |

**Random Tree Families (Bottom Block)**   The third block contains synthetic tree families designed to test whether the learned embedding space distinguishes natural-language syntax from artificially generated structures. It includes: (i) basic random trees (uniform, star, balanced, linear, and Markov models of several orders); (ii) context-free random trees sampled from grammars induced from English-EWT; and (iii) ChatGPT-generated trees, acquired by parsing with UDPipe, sentences acquired in response to SQuAD (Rajpurkar et al., 2016) prompts.

The simple synthetic families lie far from natural-language trees under both TED and our metric, confirming that natural syntax occupies a highly non-random region of tree space. Importantly, our metric produces larger and more systematic separations between natural and random trees than TED.

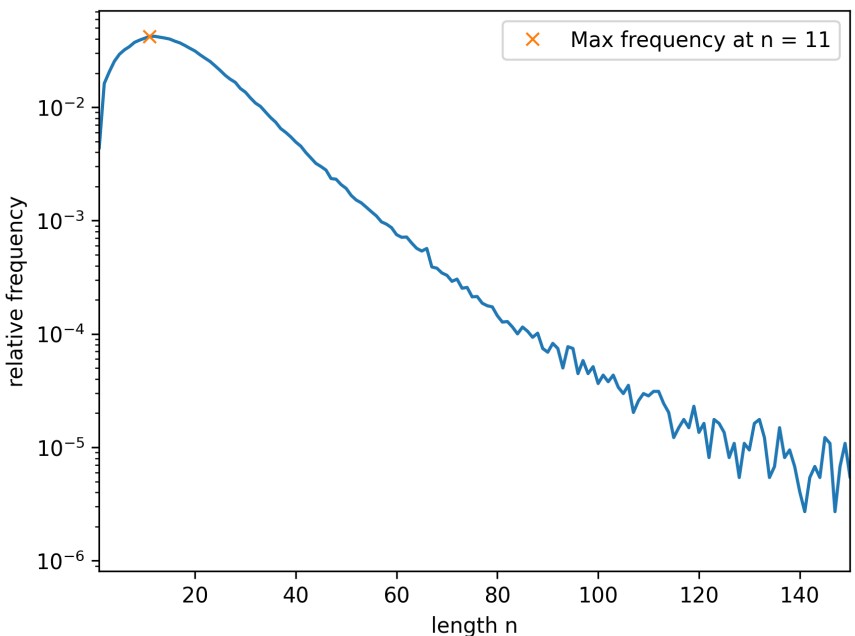

Figure 7: Distribution of sentence lengths for all natural language data we used.

Furthermore, the embedding-induced distances reveal a clear generational trend in LLM outputs: syntactic structures from ChatGPT-2/3 lie farther from gold English-EWT, while ChatGPT-4o and ChatGPT-5 produce increasingly EWT-like structures, as was more discussed in the main text.

## C   Simple Random Trees

For simple random trees, the size $n$ could in principle be unbounded. As a practical approach aligned with our downstream focus on natural-language syntax, we first sample the tree size $n$ from the empirical sentence-length distribution of UD treebanks before generating the random tree structure.

Figure 7 shows the distribution of all UD sentences we used. For a given $n$, a tree is constructed recursively by inserting the $j$th node in a tree of size $j-1$, $j \leq n$. The order of children is defined by the order of their creation.

We considered the following variations in this paper:

**Uniform:** A new node is attached uniformly randomly to one of the tree's previously generated nodes.

**Star:** All nodes share a common root node with all other nodes as its children.

**Balanced:** The complete binary tree of $n$ nodes (levels filled left to right).

**Linear:** Each node except the leaf node always has a single child. This forms a Markov tree of order 2.

**Markov:** For $j > 0$, a new node is attached to one of the $j$ previous nodes in the tree. The case of $j = 1$ is a linear tree (of order 2), as mentioned above, while the cases of $j = 2, 4, 9$ are examined in this work (Markov orders $3, 5, 10$, respectively).

After a tree is sampled, a signed ordered tree is sampled. For each parent $\rightarrow$ child edge, we sample a direction sign, either "+1" or "-1". The root has no sign. For Markov, the children's sign is always +1 by definition. For other random trees, children can have "-1" and "+1". Since natural language trees are non-projective (dependency branches do not cross any other), we sample only projective trees: for a node with $b$ children,

we sample an integer $c \in [0, b)$. As the children are already ordered when generated, $[0,c)$ has "-1" while the rest have "+1".

## D   Random Context-Free Tree Generation

Our set of context-free random trees was generated from an UD corpus, as follows. First, a dependency grammar was constructed for the corpus by examining each word $w$ and recording what other words modify it. As $w$ can be modified by words before or after itself, and because we are using a signed ordered tree, the sets of modifying words *before* and *after* $w$ are recorded with their frequency counts, together with the distribution of the numbers of modifiers. The resulting sets of modifier-modified relations before and after $w$ are denoted as $Gb$ and $Ga$, respectively. Furthermore, all main predicates are collected from the corpus together with their frequencies; the resulting set is denoted as $Gh$.

From this grammar, trees are generatively sampled. A main predicate is sampled from $Gh$ in proportion to its frequency. Then, a random sentence is generated recursively by using a function $F$, starting with the main predicate $w$ as the target word. For every target word $w$, from each of its $Gb$ and $Ga$, $F$ first samples the number of modifiers; then, it samples the corresponding number of modifiers in proportion to the frequency of the number of occurrences. For each modifier generated in this way as a new target, the function $F$ is called recursively.

In the grammar construction, $Gb$ and $Ga$ are empty for many words that occurred in the tree without a modifier. The recursive procedure thus terminates when a zero is sampled as the number of modifiers. In other words, our sample procedure terminates without introducing any arbitrary setting such as a stop probability.

## E   Training Sample Size $K$

We examine how the learned embedding space over discrete tree structures stabilizes as the *training sample size $K$* increases, holding all other conditions fixed (inputs, model architecture, optimizer, and evaluation settings $S = 1000$ and $I = 10$). For each value of $K$, we train a model from scratch, embed out-of-sample trees, and compute the Wasserstein (Sinkhorn) distance between the resulting sets of tree embeddings.

We evaluate two scenarios:

- **Same-set stability (EWT↔EWT), S1** of section 6.1. Both sets consist of samples from English-EWT. This measures whether the learned space converges when the underlying distribution is fixed.

- **Multi-set stability (EWT↔Others), S2** of section 6.1. One set is sampled from English-EWT, and the other from Japanese-GSD, Korean-Kaist, Icelandic-IcePaHC, and Hindi-HDTB. For each $K$, we compute the resulting cross-dataset distances to assess how the metric stabilizes across heterogeneous tree distributions.

**Training sizes.**   We sweep $K \in \{500, 1500\} \cup \{2000, 3000, \dots, 12000\}$ for both scenarios.

**Findings.**   (*i*) **Same-set.** The Wasserstein distance increases rapidly as $K$ grows and plateaus around $K \approx 9,000$, with small standard deviations beyond this point. (*ii*) **Multi-set.** For $K < 8,000$, both the mean distance and its standard deviation fluctuate noticeably; once $K \geq 10,000$, both the distances and the error bars stabilize. Given total corpus size and computational budget, we adopt $K = 10,000$ as a reliable setting.

## F   Sample Size $S$

We investigate the effect of evaluation sample size $S$ by measuring stability of the Wasserstein distance. Using the five-language setup (**S2** of section 6.1: English-EWT, Hindi-HDTB, Japanese-GSD, Korean-Kaist, Icelandic-IcePaHC), we sweep $S \in \{250, 500, \dots, 2000\}$ while keeping all other settings unchanged.

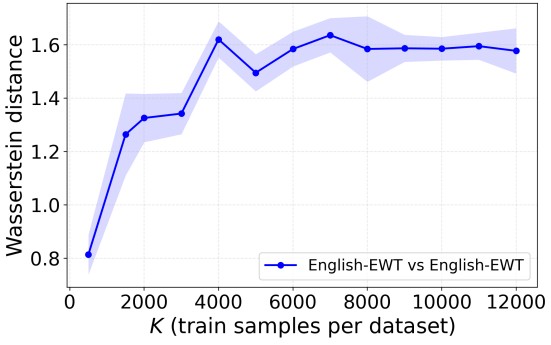 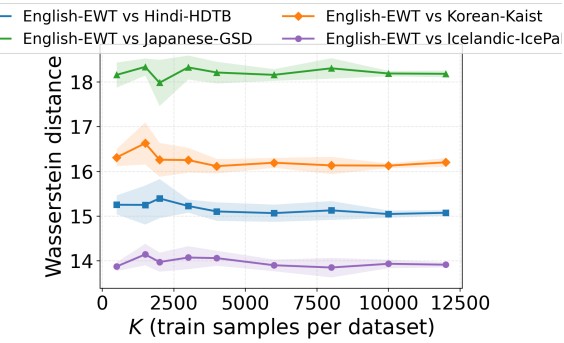

Figure 8: **Convergence of the learned embedding geometry with respect to the number of training trees $K$. Left:** Same-set stability (English-EWT $\leftrightarrow$ English-EWT). Distances rise rapidly at small $K$ and plateau near $K \approx 8000$. **Right:** Multi-set stability (English-EWT vs. Hindi-HDTB, Japanese-GSD, Korean-Kaist, Icelandic-IcePaHC). Distances exhibit larger variability when $K < 8000$ and stabilize for $K \geq 10{,}000$. Error bars denote mean±std over $I = 10$ trials.

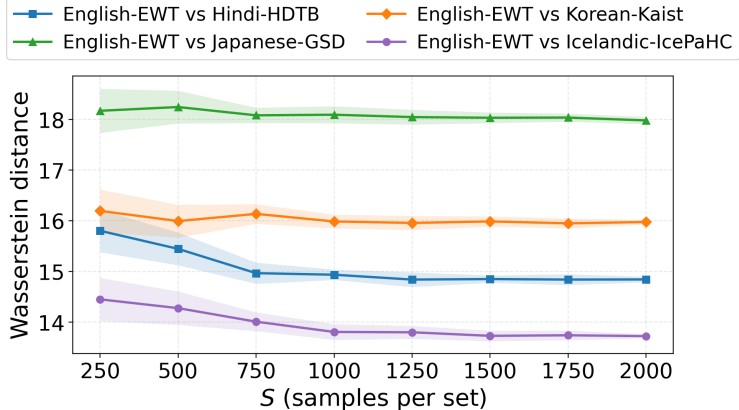

Figure 9: **Effect of evaluation sample size $S$ on Wasserstein distance estimates.** Variability is high at small $S$, but distances stabilize and show low variance once $S \approx 1000$. Error bars show mean±std over $I = 10$ trials.

The distance stabilizes and exhibits low variance when $S \geq 1000$. Balancing computational cost and reliability, we therefore use $S = 1000$ throughout the main experiments.

## G  Selecting $(H, \alpha)$ via a Joint Sweep

We jointly tune the embedding dimension $H$ and the language–prediction weight $\alpha$ on the five-language setup described in §6.1. We evaluate the three structural prediction tasks used throughout the paper: node position (ID, short for Seq-ID in the main paper), parent position (PID, for Parent-ID), and language (LANG, for Tree Class). We report mean $\pm$ std over $I = 10$ runs.

**Grid and selection rule.**  We sweep $H \in \{64, 96, 128, 192, 256\}$ and $\alpha \in \{0.0, 0.1, 0.3, 0.5, 0.6, 0.8, 1.0\}$, i.e. $5 \times 7 = 35$ parameter configurations.

To avoid over-optimizing a single task, we considered joint weighted score as follows:

$$\text{Score} = w_1 \cdot \text{ID} + w_2 \cdot \text{PID} + w_3 \cdot \text{LANG}.$$

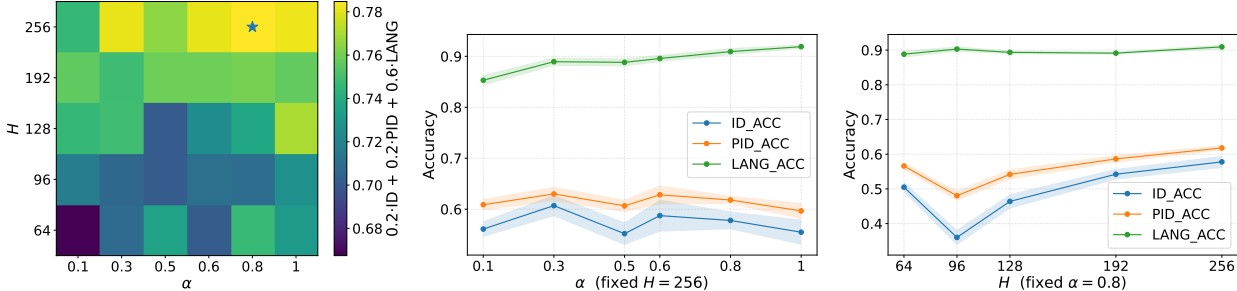

Figure 10: **Joint sweep over** $(H, \alpha)$**.** Left: Weighted score heatmap when $w_1 = w_2 = 0.2$, $w_3 = 0.6$, with a marked optimum at $(256, 0.8)$. Top-right: Accuracy profiles as $\alpha$ varies at fixed $H{=}256$. Bottom: Accuracy profiles as $H$ varies at fixed $\alpha{=}0.8$. Error bands denote mean±std when available.

**Findings.** For each of the seven alternative weight settings, we produced a heatmap as shown in figure 10 (left). The best score was obtained at

$$H = 256, \qquad \alpha = 0.8,$$

which performed best across multiple weight settings. We therefore adopt this configuration for all main experiments. The accuracy behavior with respect to $\alpha$ and $H$ is shown in the middle and right figures.

## H    t-SNE Settings

For visualization of the embedding space, a small number of points appear as outliers far from the main clusters. To improve clarity while preserving underlying cluster structure, we apply a light outlier-removal step before running t-SNE.

After $z$-score normalization in embedding space, each point is scored by its mean $k$-nearest-neighbor distance ($k{=}10$). We remove the top 1% highest-scoring points within each class, which suppresses extreme outliers while maintaining internal geometry.

t-SNE is then run on the pairwise embedding-induced distance, as mentioned in the main text. Finally, excess whitespace in the rendered plot is trimmed so that the main clusters occupy the figure area without distraction.

## I    Prediction Model Baselines

### I.1    TreeLSTM (Tai et al., 2015)

**What it is.** The Child-Sum TreeLSTM extends an LSTM from a chain to a dependency tree by composing each parent state from its children (e.g., summing child hidden states and applying LSTM gates).

**How it differs from vanilla LSTM.** A standard LSTM propagates along a single sequence; TreeLSTM performs bottom-up composition over a tree, combining multiple children at each parent.

**Application to our evaluation of section 6.1.** We run TreeLSTM bottom-up on the dependency tree and obtain the sentence representation by mean pooling over token states, matching our main model. Training, objectives, and all optimization settings are identical to the main model; only the encoder is replaced.

### I.2    Baseline II: Tree Transformer (Wang et al., 2019)

**What it is.** A Tree Transformer modifies self-attention to prefer token pairs that are close in the tree, by reweighting attention with a fixed tree-based prior matrix. Concretely, at each layer it multiplies the standard attention weights by a prior $C \in [0, 1]^{n \times n}$, where $C_{ij}$ is larger when tokens $i$ and $j$ lie in the same subtree; setting $C$ to all ones recovers a vanilla Transformer.

**How it differs from vanilla Transformer.** Vanilla attention depends only on content similarity, while the Tree Transformer injects an explicit tree bias via $C$.

**Application to our evaluation of section 6.1.** We keep the same inputs, pooling, heads, and training protocol as our main model, and replace only the encoder with the Tree Transformer.

### I.3 Baseline III: Graph Convolutional Network (GCN)

**What it is.** A graph convolutional network (GCN) produces token representations by repeatedly aggregating neighbor information on the tree graph (message passing) (Kipf & Welling, 2017), followed by a simple pooling step to obtain a sentence representation.

**How it differs from our model.** Our main encoder operates on a linearized tree and can model long-range interactions via self-attention, whereas a GCN operates directly on the graph and aggregates information locally over a small number of hops.

**Inputs and construction.** We convert each dependency tree into an undirected token graph by adding an edge for each head–dependent arc. Node features are the same structure-only template as in the main model. Pooling, heads, and training protocol are identical to the main model.

### I.4 Baseline IV: Graph2Vec

An unsupervised method that learns a fixed-dimensional embedding for each graph, inspired by Doc2Vec (Narayanan et al., 2017). Given a collection of graphs $\{G_i\}_{i=1}^{M}$, Graph2Vec represents each graph $G_i$ as a multiset of rooted subgraphs ("graph words") extracted by Weisfeiler–Lehman (WL) relabeling for a fixed number of iterations. It then trains a skip-gram / Doc2Vec-style objective so that the graph embedding vector $\mathbf{g}_i \in \mathbb{R}^d$ is predictive of the subgraph tokens that occur in $G_i$.

**How it differs from supervised baselines.** Unlike prediction-based baselines such as GCN or TreeLSTM, Graph2Vec uses no labels and no task-specific supervision. It learns graph-level representations purely from recurring structural patterns, providing an unsupervised reference for how much structure is recoverable without training objectives tied to our model.

## J  Baselines of Tree Metrics

For comparison with our embedding-induced distance learned through structural prediction, we evaluate several classical and widely used tree-metric baselines. Below we summarize the configurations used in our experiments, chosen to provide strong and representative baselines across discrete tree-structured data.

**TED (Tree Edit Distance):** We use the standard Zhang–Shasha (ZSS) implementation with all edit costs fixed to 1. This yields the classical form of tree edit distance commonly adopted in the tree-structure literature, and it serves as a reference point for assessing how our learned embedding space differs from traditional edit-based measures of structural similarity. For speed test only in section 5.1, we used APTED (Pawlik & Augsten, 2016).

**Tree Kernels:** We adopt the *exact* Collins–Duffy Subset Tree Kernel (SST) with decay parameter $\lambda$=0.4. Kernel values $K(x, y)$ are converted into distances using the kernel-induced metric:

$$m_{\mathrm{TK}}(x, y) = \sqrt{\max\{0,\ K(x, x) + K(y, y) - 2K(x, y)\}}.$$

This baseline captures subtree overlap rather than edit operations and thus provides a complementary perspective on structural similarity.

**Gromov–Wasserstein (GW):** We compute the GW distance using POT's `ot.gromov` implementation, with uniform node masses and shortest-path distances within each tree as the ground metric. This optimal-transport–based method compares metric spaces rather than explicit tree shapes and therefore serves as a geometry-sensitive baseline.

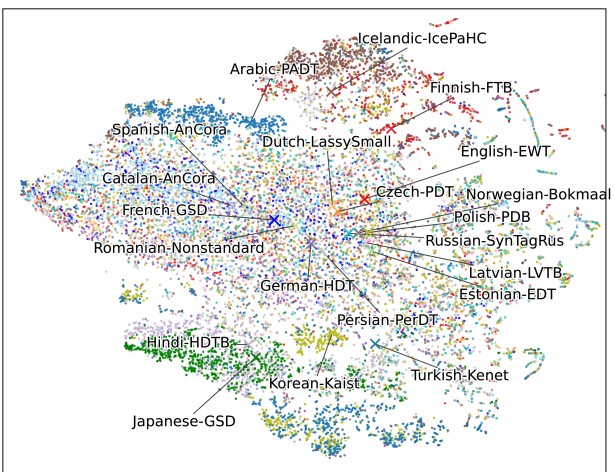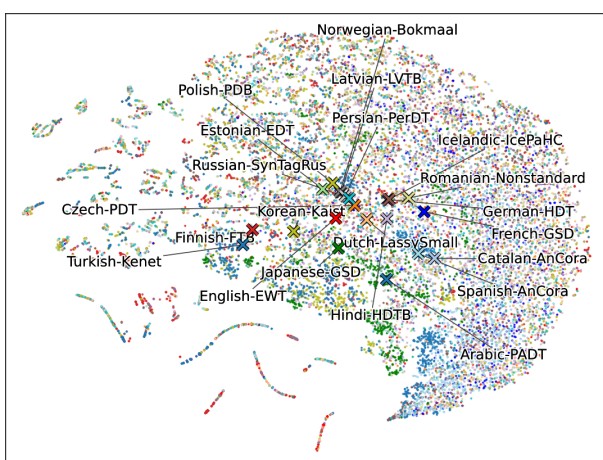

Figure 11: Point clouds of all natural-language treebanks ($S = 1000$ per language) embedded in the learned tree-structure space. **Left:** Model trained *with* language labels. **Right:** Ablated version trained *without* language labels. Centroids for each language are also plotted.

**PQ-Grams:** We use the `PyGram` library with parameters $p=2$ and $q=3$. PQ-grams provide a fast, locality-sensitive structural profile of trees and are widely used in large-scale tree comparison settings.

These baselines together span classic edit-based distances, kernel methods, optimal-transport metrics, and locality-sensitive structural approximations, providing a comprehensive suite for evaluating the effectiveness of embedding spaces learned via structural prediction.

# K  Cross-Linguistic Variation

Figure 11 (left) visualizes the induced geometry of all 21 natural-language UD treebanks under the learned *structure-only* embedding space. Even though the encoder observes no lexical information, POS tags, or morphological features, the resulting clusters align well with known typological groupings. Romance and Slavic languages form coherent neighborhoods; several SVO languages occupy the central region; and SOV languages cluster toward the bottom of the map. These patterns indicate that the embedding space, learned solely from structural prediction tasks, recovers stable cross-linguistic regularities directly from tree shape.

Importantly, the visualization also highlights structural discrepancies within the UD framework itself. For example, Japanese and Korean—often considered typologically close—form distinctly separated clusters, echoing previously reported inconsistencies in UD annotation practices for these languages (Han et al., 2020). The metric space therefore offers a quantitative, geometry-based lens for evaluating annotation conventions and potentially guiding revisions to cross-linguistic annotation standards.

Figure 11 (right) shows the ablation trained *without* language labels. While the clusters become less sharply defined, the major linguistic groupings—such as the SOV family—remain distinct. This demonstrates that much of the cross-linguistic structure emerges from purely structural prediction.

These findings suggest that a learned space of trees can serve as a powerful tool for cross-linguistic analysis, typological inquiry, and evaluation of syntactic annotation frameworks, all without relying on lexical or semantic information.

