# OpenReview forum: "Learning Embeddings for Discrete Tree-Structured Data via Structural Prediction"
_TMLR — Decision pending for TMLR_

### Review · Reviewer_9vDf · 2026-02-18

**Summary Of Contributions:**

The paper proposes a framework to map discrete, rooted, ordered trees—such as natural language dependency trees and program Abstract Syntax Trees (ASTs)—into a coherent Euclidean embedding space. Unlike traditional methods that supervise distances directly using computationally expensive metrics, this model uses self-supervised structural prediction.

**Additional Comments:**

The paper acknowledges it is dealing with "ubiquitous" discrete trees, yet the reliance on tree edit distance (TED) as the gold standard for comparison feels disconnected from modern advancements that have moved past simple parse tree comparisons

The authors define the representation using a Transformer encoder but treat the tree primarily as a linearized sequence. There is a missed opportunity to discuss how this compares to Graph Transformers that process the adjacency matrix directly without requiring a deterministic serialization.

The mechanism where Eq. 4 and 5 (predicting positions and parents) leads to a stable Euclidean geometry is not intuitively explained. The paper needs to bridge the gap between "node-level classification tasks" and "global metric space properties" more clearly.

While the paper demonstrates effectiveness on ASTs and UD treebanks , it stays within the realm of relatively simple, rooted, ordered trees. It does not address the complexities found in more modern structured learning applications, which might make the work feel less "timely" to a cutting-edge machine learning audience.

**Audience:**

No

**Audience Explanation:**

Reasons for Limited Appeal

The LLM Paradigm Shift: In the era of Large Language Models (LLMs), the focus on explicit parse trees and symbolic structures is increasingly niche. Most modern representation learning has moved toward implicit embeddings where "structure" is learned through next-token prediction rather than explicit tree topology.

Domain Maturity: As highlighted in your comments, structured representation learning is now a mature field with high-stakes applications in complex domains like protein structure prediction. Compared to those challenges, simple syntax trees or program execution trees may seem like "solved" or "over-simplified" problems.

Lack of Semantic Integration: The model explicitly ignores lexical and semantic information. While this enables domain-generalization, it limits the paper's utility for the majority of the TMLR audience who are focused on semantic understanding.

**Claims And Evidence:**

No

**Claims Explanation:**

While the authors provide empirical data showing the efficiency and stability of their proposed metric, the evidence is weakened by the use of outdated baselines and a lack of clear conceptual intuition regarding the training mechanism.

Evidence of Efficiency and Stability

- Computational Scalability: The claim that the method provides a scalable comparison function is supported by Table 2, which shows the embedding-induced distance operates in $O(H)$ time, significantly faster than the $O(n^3)$ complexity of Tree Edit Distance (TED).

- Geometric Stability: The authors provide evidence for "empirical stability" in Figure 2, demonstrating that embedding distances grow smoothly and approximately linearly with the number of local edits, unlike the "abrupt jumps" often seen in discrete combinatorial methods.
- Correlation with Classical Metrics: Despite not being supervised by TED, the learned embeddings show a strong Pearson correlation (0.75) with it, suggesting the structural tasks do capture fundamental tree topology.

Weaknesses in Evidence and Timeliness

- Outdated Baselines: As noted in your comments, the work focuses on classic problems using methodologies that may be viewed as "backwards" in the current AI landscape. The primary neural baselines—TreeLSTM (2015) and GCN (2017)—are many years old and do not reflect the current state of the art.
- Lack of Modern Graph Methods: While the paper treats trees as a specific form of graph, it fails to discuss or compare against Graph Transformers or modern structural representation learning methods commonly used today.
- Unclear Training Intuition: The submission lacks a clear explanation for why predicting parent indices and token positions should induce a coherent Euclidean space. While the results are documented, the fundamental "why" behind the effective convergence of these specific supervision signals remains opaque.

**Requested Changes:**

1. Incorporate recent works, recent 2 years.
2. Try to compare with Graph Transformers.
3. Please illustrate the motivation of the work in the era of LLM.

---

### Review · Reviewer_rjpw · 2026-04-10

**Summary Of Contributions:**

The paper describes a framework to learn embeddings of tree structures. The embedding tries to capture stability under local structural perturbations and be useful for a set of downstream tasks.

Contributions:
1. Synthetic experiments to validate alignment to TED / edit steps in uniform-random trees.
2. Usefulness to downstream tasks like predicting node index, index of parent, and tree class; identifying parser quality; Evaluating language models; code clone detection.

I would recommend to accept this paper because this area (embedding natural language trees) deserves attention from researchers. It's a less cared about niche area destroyed by the trend of LLMs. However, numerous work in psycholinguistics and others have demonstrated our cognitive bias towards understanding sequences as tree-like structures. Therefore, I find this type of work still interesting and relevant and worth publishing.

**Additional Comments:**

I have some additional questions:
1. Do you use any norm penalty on the produced embedding? The loss itself (equation 6, 7) doesn't seem to have such constraint. Further, I noticed in Figure 2, middle and right, the distance mean is different (around 0-16 for middle, but around 0-20 for right). You set H=256, what happens if you set H=512? I understand these are parametric distance models -- so we should only make comparisons within the same model. I'm only a tiny bit curious about how well the stability assumption holds across different embedding sizes.

2. Is the central point of Sec 5.2 and Figure 2 that the line closer to a straight diagonal line the better (smooth to the edit step)? If so, the right figure $d_{\theta}$ is only mildly better than Jaccard. Is my interpretation correct?

3. In Section 5.3, why is high correlation with TED desirable? Is it because for uniform random trees, TED is the optimal metric, therefore closer is better? But for natural language trees, TED is suboptimal, therefore alignment to it is worse/bad?

**Audience:**

Yes

**Audience Explanation:**

This paper worked on a relatively niche area but actually can have broad appeal to the ML community.

First of all, embedding models have been dominated by LLMs lately. It's quite refreshing to see people pushing on classical methods. The idea is similar to language model pre-training -- a predictive objective (next token prediction vs token position, parent index, and class) will give rise to an embedding space. I don't find it surprising at all, but it's still good to see it exist and being experimentally validated.

Second, trees exist in many disciplines -- phylogenetics, medical codes (ICD-9/10, SNOMED-CT), file systems, etc. A lot of the subfields of CS rely on computational linguists to propose methods to encode trees. Pushing on this direction is quite important. Publishing this work will encourage follow-up works that further this field.

Third, learned metric space itself is topic that connects mathematics and computation -- this paper can encourage more collaboration between researchers in different disciplines.

**Broader Impact Concerns:**

No need.

**Claims And Evidence:**

Yes

**Claims Explanation:**

The contributions are well supported. The strength of this framework is quite clear:
1. Stable under local perturbations
2. Very fast to encode (although, all neural methods like TreeLSTM, TreeTransformer, GCN, Graph2Vec would be fast) (it's not compared in Table 2).
3. High correlation with TED in random tree corpus (this is impressive)
4. General good performance in downstream tasks.

**Requested Changes:**

I might have some minor (optional) requests:
1. Since you guys didn't have a norm penalty (see addional comments), can you run an experiment with H=512, H=1024 to see if the local perturbation would still hold?
2. This type of method -- the embedding time is fast (Table 2), but training time might not. Can you also report how long it took for you to train the method? I'm curious how it compares to TED and APTED when you factor in training. I bet there is a chance your method is actually slower or at least on-par with TED. TED requires no training at all. Please report this in the appendix even if the paper is accepted.

---

### Review · Reviewer_54H9 · 2026-04-24

**Summary Of Contributions:**

This paper proposes a learned embedding-based method for tree-structure distance measurement and other downstream applications. Based on a transformer encoder and a learnable vector (named as TRV), the authors devise tasks to predict the node position, the parent position, and the whole structure’s type.

- Based on these structure-aware prediction tasks, the proposed method reaches a high correlation with TED.
- The method is intuitive and simple, while it significantly accelerate the tree distance obtaining process especially on large-scale datasets with very large trees.
- Calculating distances via embeddings is stable and may resolve the mutation/re-ordering problems in TED.

**Audience:**

Yes

**Audience Explanation:**

Despite the methodologies and experimental issues, the core idea about learning valid representations for structure-aware distance calculation is novel, and the method may be used for downstream tasks especially when the tree is very large with many nodes.

**Broader Impact Concerns:**

There is no ethical concerns for this paper.

**Claims And Evidence:**

No

**Claims Explanation:**

Some of the authors’ proposed claims are not well supported by current evidence.

1. The biggest concern is about the evaluation strategy. The paper explicitly rejects TED as a training signal because it does not reflect the regularities that arise from local order relations and is discrete combinatorial quantity with abrupt jumps. However, some of the performance are evaluated by presenting a correlation with TED. If TED is not a valid signal for training, why it is a suitable method for evaluation? If it is a valid evaluation method, why not training models with TED supervisions?
2. The detailed ablation study about the stability claims are absent. There are no fair comparisons on cases where TED encounter abrupt changes while current method does not.
3. There lacks of more strict comparisons with other neural baselines. e.g. What if we integrate TRV into tree transformer with the same prediction tasks?
4. The efficiency analysis is kind of tricky. The implementation details of other baselines are missing. Maybe it would be much faster with c/cpp/rust implementations for TED and its variants? Besides, there should be an end-to-end comparison besides the distance calculation only, since the majority of time for embedding-based methods would fall into the encoding stage.

**Requested Changes:**

1. The paper should provide a clear justification for why TED is suitable for evaluation but not for training. If not, the claim about current proposed method is superior to the others may be not hold.
2. Maybe we should add the other neural baselines for the code clone retrieval task in section 7.
3. Adding more ablation studies about the training tasks.
4. Adding direct stability comparison with TED and its variants on curated evaluation subsets (mutation and order-sensitive conditions and different tree structure types with different tree sizes).
5. Adding more details and the e2e latency results when analyzing efficiency.
6. Presenting the results of the number of nodes vs. TED correlation to check the effectiveness when the number of node scales.